# Designing Functionally Substituted Pyridine-Carbohydrazides for Potent Antibacterial and Devouring Antifungal Effect on Multidrug Resistant (MDR) Strains

**DOI:** 10.3390/molecules28010212

**Published:** 2022-12-26

**Authors:** Farooq-Ahmad Khan, Sana Yaqoob, Shujaat Ali, Nimra Tanveer, Yan Wang, Sajda Ashraf, Khwaja Ali Hasan, Shaden A. M. Khalifa, Qiyang Shou, Zaheer Ul-Haq, Zi-Hua Jiang, Hesham R. El-Seedi

**Affiliations:** 1Third World Center for Science and Technology, International Center for Chemical and Biological Sciences, University of Karachi, Karachi 75270, Pakistan; 2H.E.J. Research Institute of Chemistry, International Center for Chemical and Biological Sciences, University of Karachi, Karachi 75270, Pakistan; 3Molecular and Structural Biology Research Laboratory, Department of Biochemistry, University of Karachi, Karachi 75270, Pakistan; 4Department of Molecular Biosciences, The Wenner-Gren Institute, Stockholm University, S-106 91 Stockholm, Sweden; 5Second Clinical Medical College, Zhejiang Chinese Medical University, Hangzhou 310053, China; 6Dr. Panjwani Center for Molecular Medicine and Drug Research, International Center for Chemical and Biological Sciences, University of Karachi, Karachi 75270, Pakistan; 7Department of Chemistry, Lakehead University, 955 Oliver Road, Thunder Bay, Ontario P7B 5E1, Canada; 8International Research Center for Food Nutrition and Safety, Jiangsu University, Zhenjiang 212013, China; 9Pharmacognosy Group, Department of Pharmaceutical Biosciences, BMC, Uppsala University, Box 591, SE-751 24 Uppsala, Sweden; 10International Joint Research Laboratory of Intelligent Agriculture and Agri-Products Processing (Jiangsu University), Jiangsu Education Department, Nanjing 210024, China

**Keywords:** multidrug resistance, MDR strains, antibiotics resistance, antimicrobial, pyridine

## Abstract

The emergence of multidrug-resistant (MDR) pathogens and the gradual depletion of available antibiotics have exacerbated the need for novel antimicrobial agents with minimal toxicity. Herein, we report functionally substituted pyridine carbohydrazide with remarkable antimicrobial effect on multi-drug resistant strains. In the series, compound 6 had potent activity against four MDR strains of *Candida* spp., with minimum inhibitory concentration (MIC) values being in the range of 16–24 µg/mL and percentage inhibition up to 92.57%, which was exceptional when compared to broad-spectrum antifungal drug fluconazole (MIC = 20 µg/mL, 81.88% inhibition). Substitution of the octyl chain in 6 with a shorter butyl chain resulted in a significant anti-bacterial effect of 4 against *Pseudomonas aeruginosa* (ATCC 27853), the MIC value being 2-fold superior to the standard combination of ampicillin/cloxacillin. Time-kill kinetics assays were used to discern the efficacy and pharmacodynamics of the potent compounds. Further, hemolysis tests confirmed that both compounds had better safety profiles than the standard drugs. Besides, molecular docking simulations were used to further explore their mode of interaction with target proteins. Overall results suggest that these compounds have the potential to become promising antimicrobial drugs against MDR strains.

## 1. Introduction

Multi-drug resistant (MDR) pathogens have emerged as a major challenge of the 21st century because life-threatening infections are gradually limiting our therapeutic options, especially for debilitated individuals and immunocompromised patients [1,2]. Potential health benefits of conventional drugs are in jeopardy since antibiotics are losing their efficacy against infections, as they frequently cause toxic reactions from prolonged use and misuse, thus instigating the emergence of drug-resistant strains. The problem caused by fatal superbugs currently represents one of the main areas of unmet medical needs [3]. In this alarming situation, it is essential to investigate novel drugs with lesser resistance. Nevertheless, the development of new antibiotics is rolling substantially slower than our growing demand in the war against microbial infections [4]. Hence, there is an urgent need to design novel treatment regimens and explore effective therapeutic options to cure the infections caused by problematic MDR strains [5]. 

Heterocycles are ubiquitous and are widely found in nature as alkaloids, hormones, and metabolites. Pyridine is particularly an important nitrogen-containing heteroarene with great utility in medicinal chemistry [6,7]. More than 50 drugs in the database of US-FDA stem from pyridine, including pyridostigmine for myasthenia gravis, dexamethasone isonicotinate for inflammation and allergy, piroxicam for arthritis, crizotinib for cancer, tacrine for Alzheimer’s, abiraterone acetate for prostate cancer, delavirdine for HIV/AIDS, and many more [6]. For the treatment of bacterial and fungal infections, many FDA-approved drugs also contain pyridine scaffolds (Figure 1), such as ceftazidime, mainly for *Pseudomonas aeruginosa*, tedizolid, ozenoxacin and prestinamycin for *Staphylococcus aureus* infections, prothionamide for *Mycobacterium tuberculosis* and ethionamide for MDR *Mycobacterium tuberculosis*, nalidixic acid for *Escherichia coli*, *Proteus mirables* and *Pseudomonas aeruginosa*, trichodin A for *Candida albicans* and many more.

The pyridine ring offers favorable intermolecular interactions with drug targets through 𝜋-𝜋 stacking, hydrogen bonding, and metal coordination, and it is a widely used structural element in drug design. Therefore, this scaffold is an integral part of numerous broad-spectrum antibiotics (Figure 2), including rifaximin, to treat traveler’s diarrhea by binding with RNA polymerase enzyme of bacteria, ceftaroline fosamil–against many Gram-positive bacteria, especially methicillin-resistant *Staphylococcus aureus* (MRSA), nikkomycin which binds with fungal cell wall to inhibit chitin synthase.

Hydrazides are a class of organic compounds having the RNHNH_2_ formula, where R is either phosphoryl (R_2_P(O)^−^), sulfonyl (RSO_2_^−^), or acyl (RCO-). They are also useful building blocks to synthesize various bioactive compounds against microbial infections [5]. For example, oxadiazoles, triazoles amine, Schiff bases, and pyrazoles are some known derivatives of hydrazides with significant biological activities [8]. Owing to their chemical reactivity and biological activities [9], hydrazide-based compounds are useful substrates in both realms of chemical reactions and medicinal chemistry (Figure 3). They impart potent pharmacological activities due to the presence of toxophoric N-C=O linkage, thus displaying antiviral, antimicrobial, local anesthetic, anticonvulsant, antioxidant, anti-inflammatory, anticancer, and possible antimycotic properties [8,10].

The fascination with hydrazides is not simply a consequence of their derivatives with good pharmacological activity; the hydrazide moiety is used as an active functional group in numerous chemical reactions, including the synthesis of tetrazoles [11], the preparation of spiroquinazolinones [12,13], Ugi-azide reactions [12], Ugi-multicomponent reactions [14], and many more organic reactions. Furthermore, hydrazides can also be employed in chemical synthesis as organo-catalysts [15]. However, the use of hydrazides as starting materials is a challenging task due to the regio-selectivity between the competing amines (Figure 4) contained in its structure, N1 and N2, respectively [16]. While its reactivity through the N2 has largely been restricted to the production of hydrazones, it has also been studied in Michael addition reactions and cross-coupling reactions. A brief review of the literature reveals that in the asymmetric synthesis of organic compounds of medicinal importance, hydrazides serve the role of reagents as well as the role of chiral catalysts [9].

A number of pyridine-hydrazide-phytochemicals have recently been identified as antimicrobial agents (Figure 5). Given its biological activity and use as an anti-tuberculosis drug, pyridine-4-carbohydrazide, also known as isonicotinoyl hydrazide (INH), is one of the most well-known representatives of the hydrazide family [9]. It has been suggested that incorporating lipophilic functionalities into the framework of INH augments the biological activity of the drug by facilitating diffusion through the lipid-rich bacterial wall [17]. In order to enhance the tissue availability of the INH molecule, new lipophilic INH derivatives are emerging as potential antibacterial agents [18]. In the field of anti-tuberculosis actions, INH and its acyl-substituted heterocycles have grown significantly in prominence [19]. Quinolines are ubiquitous natural products having diverse biological activities. This pharmacophoric moiety is an integral part of a recently approved anti-TB drug TMC207. Pitta et al. reported notable activity against *M. tuberculosis* for quinoline derivatives of pyridine-4-carbohydrazide [17]. Sharipova et al. conjugated pyridine-4-hydrazide with some glycosides, which were isolated from *Stevia rebaudiana.* For these compounds, moderate activity against *M. tuberculosis* strain H_37_R_V_ was reported [20]_._ De et al. synthesized cinnamic acid derivatives of pyridine-4-carbohydrazide and found submicromolar MIC values against the same strain (H_37_Rv) [21]. Ramani et al. also found remarkable activities against *M. tuberculosis*, some compounds being more active than the standard drug. The most potent compound (*N*-(2-(4-(benzyloxy) phenyl)-4-oxo-1,3-thiazinan-3-yl) isonicotinamide) inhibited *the Mycobacterium tuberculosis* strain H37Rv with MIC of 0.12 μM and was three times more potent than INH [22]. Bhilare et al. prepared phenolic acid (gallic acid, syringic acid, and vanillic acid) tethered pyridine-4-carbohydrazides for the abrogation of drug-induced hepatotoxicity [23], whereas Nayak et al. synthesized plumbagin derivatives, which were obtained from *Plumbago zeylanica*. The compounds were tested on several *M. tuberculosis* strains, and their activities were comparable to the standard drug rifampicin [24]. Rodrigues et al. also synthesized some fatty acid derivatives with remarkable results against many strains of *M. tuberculosis*. He concluded that lipophilicity plays a crucial role in the activities of pyridine-hydrazide-type molecules [25]. Previously, Pavan et al. also arrived at the same conclusion when lipophilic moieties were incorporated into the pyridine-4-carbohydrazide framework. These compounds were tested on many bacterial strains, including *M. tuberculosis,* and their potent activities were tentatively attributed to the lipophilic nature of the molecules, which led to their enhanced penetration into host mammalian tissues and the lipid-rich bacterial wall [26]. 

It was deemed worthwhile to undertake the antimicrobial activities of the aforementioned compounds keeping in mind the significance of the preceding hetaryl nuclei and taking into account the approach to introducing pyridine moiety into hydrazide scaffolds [27]. To combat the fast-growing antibiotic resistance and negative effects, novel antimicrobial strategies are being devised [28,29,30], which include combination therapy, physicochemical approaches, treatments for resistant bacteria, mechanisms that target the enzymes or proteins responsible for inducing antimicrobial resistance, drug delivery systems and novel technologies like the CRISPR-Cas system. These various strategies may influence how multi-drug resistant (MDR) organisms are tackled in human healthcare settings. Numerous INH derivatives have demonstrated significant antimycobacterial efficacy [31]. For example, Judge et al. synthesized this type of compound to investigate their ability to fight off *Mycobacterium tuberculosis* as well as *Candida albicans*, *Bacillus subtilis*, *Staphylococcus aureus*, *Aspergillus niger*, and *Escherichia coli*. The outcome showed that pyridine-4-carbohydrazide derivatives had greater antifungal potential, notably against *C. albicans* [32]. Rasras et al. reported novel hydrazide-hydrazones of cholic acid and evaluated their antibacterial efficacy against three strains of Gram-positive and three strains of Gram-negative bacteria [33]. Rastogi and coworkers demonstrated the synergistic action of *trans*-cinnamic acid with isoniazid, rifamycin, and other well-known antimycobacterial drugs of *M. tuberculosis.* Even drug-resistant isolates had remarkable results [34]. In addition, there is a direct correlation between the rising incidence of serious infections brought on by yeast and fungus, particularly in immunocompromised or susceptible individuals, and the paucity of novel antifungal medications [35]. Invasive fungal infections are a significant cause of mortality for these patients as a result of the ongoing, rapid rise in primary and opportunistic fungal infections [36]. One of the most prevalent opportunistic fungi that cause these infections is *C. albicans* [37,38]. For them, azole is fungistatic and susceptible to resistance, whereas polyenes have substantial host toxicity; therefore, established agents do not completely address the therapeutic requirements [39]. Modified azoles, a new class of echinocandins, and pneumocandins are among the drugs currently undergoing clinical trials [40,41]. Pyridine-containing hydrazide moiety is, therefore, quite intriguing and may reveal other overlooked pharmacological functions. 

A series of pyridine-based hydrazides, which contain lipophilic chains of varying lengths, were designed and synthesized with the aim of studying their anti-microbial characteristics, followed by molecular modeling analyses to better understand the interactions of the compounds with target proteins. The results of the study will aid in discerning structural requirements in the framework of multidrug-resistant antimicrobial agents that appear to play a crucial role in their interaction with the membrane and are important in ligand binding and activation.

## 2. Results and Discussion

The rationale for the synthesis of pyridine-based hydrazides having lipophilic chains is depicted in Figure 6. A number of pyridine and hydrazide-containing derivatives have already been identified as antimicrobial agents with significant activities (Figure 4). Given its profound effects on biological activities, pyridine is one of the most extensively used heterocyclic in drug design [6]. The presence of pyridine moiety is known to enhance drug permeability and fix the protein binding issues, which results in the overall improvement of biological activities. It has been suggested that incorporating lipophilic functionalities into the pyridine carbohydrazide framework further augments the biological activity of the drug by facilitating diffusion through the lipid-rich bacterial wall [17]. In order to enhance intracellular penetration and rapid elimination via hepatic metabolism, lipophilic chains of variable lengths were, therefore, introduced at carbohydrazide moiety in our design. This approach to drug development has already been suggested as a way to either increase the basic drug action or conduct a controlled study of the pathogen’s life cycle [42].

The titled compounds were prepared by adopting a simple and straightforward method instead of using acid chlorides, which is usually employed for the preparation of this type of compound [43,44]. Fischer esterification of pyridine-3-carboxylic acid or pyridine-4-carboxylic acid in refluxing ethanol afforded their ethyl esters, which were then reacted with 80% hydrazine hydrate to get the corresponding pyridine carboxylic acid hydrazide in high yields. Acid anhydrides of varying chain length were reacted to a methanolic solution of 3- or 4-pyridine carboxylic acid hydrazide, which produced a series of pyridine-containing lipophilic hydrazides (**3**–**12**, Figure 1) in high yield (61–100%). The structures of synthesized compounds were verified by their spectroscopic data, including IR, ^1^H NMR, and MS data (Appendix A).

### 2.1. Antifungal Studies

*Candida* species ubiquitously colonize various anatomical parts of the human body, particularly the moist areas of the skin (groin, axilla, etc.), oral cavity, and genital and gastrointestinal tracts [45]. Among the fungal pathogens, more than twenty species of *Candida* cause fungal infection in susceptible and immunocompromised patients. *C. albicans* is the vibrant pathogen responsible for most nosocomial opportunistic infections. In addition, drug-resistant *C. glabrata* is an emerging pathogen considered as a new public health threat globally [45,46]. Currently, the treatment of systemic and invasive candidiasis has become an utmost challenge to scientists and clinicians. Therefore, it is urgently needed to discover new therapeutic agents for the treatment of invasive candidiasis by exploring the therapeutic potential of bioactive compounds and the assessment of drug targets through in vitro and in silico studies [46,47]. In this study, the synthesized hydrazide-based compounds, grouped into two series (**3**–**7**) and (**8**–**12**), were explored for their antifungal potential against the fluconazole-resistant *Candida* species. Table 1 reveals that compounds **5**–**7** and **10** had remarkable anti-*C. glabrata*, *C. parapsilosis*, and *C. albicans* potential in disk diffusion assay. Their zone of inhibitions (ZOI) was found to be in the range of 7–19 mm at a concentration of 10 µg/mL of these compounds. The rest of the compounds did not show ZOI comparable to fluconazole at this concentration. Therefore, only compounds **5**–**7** and **10** proceeded for further MIC and % inhibition evaluation. In the case of *C. glabrata* and *C. parapsilosis*, these compounds completely restrained their growth as indicated by the growth inhibitory indices, including minimum inhibitory concentration (MIC), percentage inhibition, clear zones (*) and antifungal index (AFI), respectively. 

The antifungal index (AFI, relative inhibition index) provides better discrimination of fungicidal effectiveness of synthetic compounds that differs in relative diffusion rates and the residual activity concerning inhibition zones tested through the Kirby-Bauer test. Therefore, assessing the AFI of the tested compound provides insight into their diffusion rates in an agar media and yeast-drug interactions in an aerobic environment. Literature data suggest that the AFI of compound **6** was comparable to Flucytosine in the case of *C. albicans* when fluconazole was employed as a positive control using the Kirby-Bauer test [48]. In our case, the compounds (Table 1) having AFI >1 were considered more effective than fluconazole (FLZ) in the disk diffusion assay, which was performed in aerobic conditions. The broth dilution method provided a microaerophilic environment for MIC determination and percentage inhibition. The MICs were in the range of 16 to 24 µg/mL for *C. glabrata* and *C. parapsilosis*; however, compound **6** had significantly better MIC value, i.e., 16 µg/mL and maximum growth inhibition of 92.57%, and the highest AFI for *C. glabrata* in 24 h (Table 1). Furthermore, the suicidal activity of these compounds varies with the assay method, the presence or absence of oxygen, and the species. For instance, more than 24 µg/mL of **7** could inhibit 68.78 % growth of *C. glabrata* in broth dilution. Whereas 10 µg/mL of **7** produced a clear zone of inhibition (25 mm) and AFI being 1.10 in an aerobic environment on nitrogen base agar. Similarly, the susceptibility of *C. parapsilosis* to **7** increased, which needed 16 µg/mL to inhibit more than 80% of the growth; however, its AFI was 0.87 only. The lowest MIC value was observed for compound **5**, which could inhibit 91% cell division of *C. parapsilosis*, compared to FLZ, having 78.31% growth inhibition at the concentration of 24 µg/mL. Among *Candida* species, *C. glabrata* ATCC 2001 was susceptible to 20 µg/mL of FLZ, exhibiting more than 81% growth inhibition in 24 h. Decrease in susceptibly (% inhibition) to FLZ (>24 µg/mL) among *C. parapsilosis* ATCC 22019 (78.31%), *C. albicans* ATCC 36082 (71.21%), and *C. albicans* CL1 (42.19%) was observed in this study. *C. albicans* is a potential pathogen that is responsible for more than eighty percent of infections in humans. Although *C. albicans* CL1 showed significant resistance to FLZ (42.19% inhibition at >24 µg/mL), compounds **5**–**7** and **10** exhibited higher % growth inhibition against this strain. In brief, the effective susceptibility profile reveals the distinct mechanistic or modulatory metabolic activity of these compounds in *C. albicans*, *C. glabrata*, and *C. parapsilosis* compared to fluconazole. Figure 7B demonstrates the time-kill kinetics of compound **6**. The compound had promising fungicidal activities at 16 µg/mL (Table 1), revealing more than 90% inhibition of *C. glabrata* ATCC 2001 in broth dilution method and clear zone (*) of inhibition via disk diffusion assay. Therefore, this compound was subject to evaluate the minimum time required to arrest the growth of *C. glabrata* ATCC 2001. Figure 5 illustrates an immediate decrease in the growth rate evident from the steep decline in the logarithmic growth of *C. glabrata* ATCC 2001 using the concentration of the compound at 0.5×, 1× and 2× of its MIC when compared to fluconazole (≥32 µg/mL) that required longer time (12 to 48 h) to reduce ≥3 log CFU/mL. The fungicidal effect of **6** initially appeared within four hours of the compound treatment. The observed results of the time-kill assay were complimented by placing the inoculums onto SDA agar plates, which revealed no growth after 48 h of the treatments.

Compound or drug-induced, toxic hemolysis or a consequence of hypersensitivity to compounds and their metabolites because of immunological reactions is rare but may cause serious toxicity liability. Although toxic hemolysis may induce at sufficiently high concentrations of drugs, however, some synthetic compounds may cause the lysis of red blood cells (RBCs) at lower doses given to individuals who are genetically predisposed to hemolysis. According to US FDA, in vitro hemolysis study should be performed to determine the toxicity of compounds at the intended concentration for administration [49]. Determining the ex vivo membranolytic potential of drugs and synthetic compounds using isolated and washed RBCs (vulnerable cells) at physiological pH is a rapid method for the initial assessment of cellular toxicity. The compounds with increasing hydrophobicity may exhibit a membrane solubilization effect, similar to detergents, and are considered toxic to mammalian cells [50,51]. Therefore, rationalizing in vitro membranolytic activity of pyridine-containing lipophilic hydrazides via hemolytic assay is crucially important to justify fungicidal and antibacterial potential with no or limited cellular toxicity to human RBCs. 

In Figure 7A, compound 6 at 2X of MIC concentration (32 μg/mL) revealed a mild increase, i.e., 2.8% lysis of red blood cells (RBCs) after the incubation period of 30 h as compared to fluconazole. In contrast, 5% DMSO was responsible for 5–27% lysis of RBCs during the assay period. However, triton X100 showed cooperative lysis behavior and lysed 100% RBCs for 30 min of incubation time.

### 2.2. Antibacterial Studies

In 1929, Alexander Fleming opened the modern era of antibiotics with the serendipitous discovery of penicillin. In human history, antimicrobial agents have been used to control ailments caused by microbial organisms, such as protozoa, viruses, fungi, and bacteria. According to World Health Organization (WHO), bacterial infection is one of the major causes of mortality in the world. The prevalence of multidrug-resistant Gram +ve and Gram −ve pathogens toward broad spectral antibiotics has reached an alarming level, and it poses a serious threat to the global population [52]. The emergence of MDR pathogens imparts challenges to clinicians and healthcare professionals for the safe management of life-threatening infections [53]. After the advent of penicillin, various bacterial genera exhibit resistance to this drug. Therefore, the surge in the discovery of potential compounds of broad spectral bactericidal activities is a prerequisite to treating the infections caused by these pathogens [54]. Consequently, to combat the spread and limit the increased resistance towards microbial agents, we have synthesized a series of pyridine-3-carboxylic and pyridine-4-carboxylic acid hydrazides of lipophilic nature antibacterial agents, which may be proven as new lead molecules possessing enhanced antibacterial activities.

Ampicillin/cloxacillin combination is used to treat a wide range of bacterial infections and is known to have a synergistic effect; therefore, they were used in combination to exert the maximum bactericidal effect of the positive control and then compare the results with the synthesized compounds. Despite the limited use of beta-lactam (ampicillin) against the *Pseudomonas aeruginosa*, the amino-penicillins can serve as a broad spectral antimicrobial agent [55]. MIC of the synthesized compounds demonstrating the ZOI (mm) ≥ to the inhibitory zone of ampicillin/cloxacillin (at a concentration of 32 μg/mL) are shown in Table 2. Compounds **4**, **5**, **8**–**10,** and **12** inhibited the growth of *Proteus mirabilis* 212a, whereas compounds **4** and **5** inhibited the growth of *Proteus mirabilis* ATCC 12453 with MIC values ranging from 4 to 16 μg/ mL, respectively. The MICs of compound **1** were 4 μg/mL, whereas that of **3** and **7** were 8 μg/mL against a clinical isolate of *Salmonella typhi* XDR ST-CL-15. Similarly, the minimum concentration of 4 μg/mL of compounds **4** and **9** were required to inhibit the growth of *Pseudomonas aeruginosa* ATCC 27853; however, only compound **4** restrained the growth of *Pseudomonas aeruginosa* PA-01 at the concentration of 8 μg/mL. The lowest MIC value of 2 μg/mL was observed for compound **3** against the *Aeromonas hydrophila* ATCC 7966 and compound **6** against the *Staphylococcus aureus* ATCC 29213. Compounds **4** and **6** had remarkable bactericidal activity in a range of disease-causing Gram-positive and Gram-negative bacterial groups. MIC of **4** was in the range of 4 to 16 μg/mL against the clinical and ATCC strains of *P. aeruginosa*, *P. mirabilis*, and *S. aureus*. Similarly, compound **6** significantly inhibited the growth of *S. aureus* ATCC 29213 at 2 μg/mL, followed by *Enterococcus faecalis* ATCC 29212, *A. hydrophila* ATCC 7966, and *S. typhi* XDR ST-CL-15. Compound **4** exerted suicidal effects at a concentration 2 to 4-fold lower than ampicillin/cloxacillin (MIC 16 to ≥32 μg/mL) onto five bacterial strains. Therefore, the time-dependent killing of compound **4** against *P. aeruginosa* PA01 and *S. aureus* ATCC 29213 was conducted at the concentrations half of MIC (0.5X), MIC (1X), and twice of MIC (2X), respectively (Figure 8). 

The data presented in Figure 8 refers to the concentration and time-dependent killing behavior of compound 4 against *P. aeruginosa* PA01 and *S. aureus* ATCC 29213. Results shows that a rise in concentration 0.5X–2X of compound 4 decreases the colony-forming units (≥3 log CFU/mL) of *P. aeruginosa* PA01 as compared to ampicillin/cloxacillin and untreated cells during 24 h (Figure 8A). Furthermore, compound 4 at 0.5X MIC (4 μg/mL) killed the *P. aeruginosa* PA01 by decreasing the cell count to a similar level as that caused by the co-amoxicillin at MIC (32 μg/mL). In contrast to the above-mentioned results, there is no significant time-dependent decrease in the log growth of *S. aureus* ATCC 29213 at 0.5X MIC of compound 4 (0.5X MIC), see Figure 8B. The killing rate of *S. aureus* ATCC 29213 appears slower than that of *P. aeruginosa* PA01 caused by four at the same concentration. In addition, 8 μg/mL of 4 produced similar suicidal effects as did 16 μg/mL of ampicillin/cloxacillin. The observed difference in time required to kill *P. aeruginosa* PA01 and *S. aureus* ATCC 29213 at 0.5X–2X of MIC for 4 is primarily due to the difference in their peptidoglycan thickness and the presence of an outer membrane in two groups of bacteria [54]. The *P. aeruginosa* PA01 possesses a 1.5–10 nm thick cell wall and 40% more lipid content as compared to the 20–80 nm thick cell wall of *S. aureus* ATCC 29213 with lower lipid content of membrane. The exerted bactericidal effects of 4 tempted as a cell membrane interfering molecule that might cause membrane misalignment in Gram-negative pathogens more effectively within a shorter period of time [54]. 

In Figure 9, compound **4** at 2X of MIC concentration (16 μg/mL) revealed no lysis of RBCs membrane during the assay period as compared to the triton X-100, SDS, and DMSO. However, SDS showed cooperative sigmoidal lysis behavior and lysed the RBCs within 30 min of incubation time. 

### 2.3. Computational Studies

In order to reveal the binding mode and inhibition mechanism of the newly synthesized pyridine-containing lipophilic hydrazides, molecular docking was performed against the well-established targets: CYP51 of *Candida glabrata* (PDB ID 5JLC) as fungal target and topoisomerase IV (PDB ID 3FV5) as bacterial target [56]. Topoisomerases IV are good antibacterial targets for several reasons: (1) they are proteins essential for bacterial viability that are involved in bacterial DNA replication, (2) they are essential components of all bacteria, (3) inhibition of their function in bacteria usually leads to a bactericidal (versus bacteriostatic) event. By considering these points, we performed the docking against topoisomerase IV, which will possibly affect many of the proteins, including the proteins involved in the maintenance of the membrane structure [57]. The potential of the newly synthesized hydrazide derivatives was measured in terms of their dock scores using the molecular docking approach. These scores present the actual strength of the non-covalent interactions between the molecules and the target protein. The more negative score corresponds to the higher binding affinity of the molecule towards its target protein. In this study, compound 4, the most potent antibacterial agent against topoisomerase IV (PDB ID 3FV5), displayed a docking score of −6.4 kcal/mol. Similarly, compounds 5–7 and 10 being the potential antifungal agents, had significant docking scores of −6.2, −6.6, −7.1, and −6.3 kcal/mol, respectively. 

The docking investigations indicate that the newly synthesized hydrazide derivatives can fit into the putative active site of CYP51 by mutual interactions, including heme coordination, hydrophobic, hydrogen bonding, and π- π stacking interactions (Table 3). In the case of 5, one of the nitrogens in the hydrazide group formed strong H-bond interaction with the Met512 at a distance of 2.0 Å (Figure 10a). While the long hydrophobic chain and the aromatic ring of the compound could interact with the nearest hydrophobic residues Leu130, Thr131, Phe237, and Phe242, the system was further stabilized by π-π stacking interaction with Tyr127 in the protein cavity. Compound 6 had two hydrogen bonds with the backbone amides of His382 and Ser383 at a distance of 2.8 and 3.0 Å, respectively (Figure 10b). Further anchorage was provided by a face-to-face interaction between the compound and His382. In the case of 7, the nitrogen of the pyridine ring was able to donate its lone pair to His382 and Ser383 to mediate hydrophilic interactions at a distance of 2.8 and 3.1 Å (Figure 10c). Another hydrogen bond was observed between the nitrogen of the hydrazide group and the side chain of Ty127 at a distance of 3.1 Å. The protein−ligand contacts were further stabilized by hydrophobic interactions between the compound and the pocket residues Tyr127, Thr131, and Leu130. In the case of antifungal agent 10, the pyridine ring had π-cation interaction with the aromatic ring of His382 (Figure 10d). However, the NH and CO groups of the compound were involved in mediating hydrogen bond interaction with residue His382 and Ser383 at the distance of 3.2, 2.1, and 2.3 Å, respectively. Furthermore, these protein-ligand interactions were stabilized by hydrophobic interactions with the crucial residues Tyr127, Thr131, and Leu130.

Figure 11 depicts the binding mode of compound **4** in antibacterial target protein topoisomerase IV. The linker carbonyl and nitrogen of the hydrazide group could mediate hydrogen bond interaction with Arg72 and Thr163 at a distance of 3.4 and 3.3 Å, respectively. Another strong hydrogen bond interaction was observed between the nitrogen of the pyridine ring and the side chain of Arg132 at a distance of 2.9 Å. Additionally, the hydrophobic chain and the pyridine ring of the compound had hydrophobic interactions with Asn42 and Pro75, as well as π-cation interaction with Arg72.

## 3. Materials and Methods

Unless otherwise stated, the various starting materials for the reaction have been procured from various commercial suppliers and used without further purification. For synthesis, analytical- or HPLC-grade solvents were used. Reaction monitoring and observations were done using TLC analysis with silica gel 60 F_254_ aluminum-coated plates. Chemical shifts (δ) are expressed in ppm, while coupling constants (J) are reported in Hz. Their multiplicities are denoted by the letters singlet (s), doublet (d), triplet (t), quartet (q), multiplet (m), broad (br), apparent (app), and various combinations of these. To determine and validate the *m/z* ratio, mass spectra were captured using a mass spectrometer with model no. JEOL 600H-1. Infrared spectra were recorded using KBr discs in a range of 4000–500 cm. The wave number is expressed in cm^−1^.

### 3.1. General Procedure for the Synthesis of Pyridine-3-Carbohydrazides and Pyridine-4-Carbohydrazides Containing Lipophilic Chains 

The synthetic route for the preparation of pyridine carbohydrazides is outlined in Figure 1. Briefly, ethyl esters of pyridine-3-carboxylic acid and pyridine-4-carboxylic acid were prepared by adding a catalytic amount of sulfuric acid into refluxing ethanolic solutions of the carboxylic acids. The progress of the reaction was monitored with TLC. After overnight reflux, the reaction mixture was cooled and neutralized with 10% sodium carbonate solution, followed by DCM extraction of the esterified product in quantitative yield. Ethyl esters of pyridine carboxylic acids were then refluxed overnight with hydrazine hydrate (80%). Then crushed ice was added to the reaction mixture, followed by washing with cold water to give pyridine carboxylic acid hydrazides. Further purifications were done by recrystallization in ethanol. To synthesize the lipophilic compounds, these hydrazides were dissolved in methanol, and then one equivalent of different acid anhydrides was slowly added to the reaction mixture. TLC was used to monitor the reaction progress. Further purifications were done with a silica gel column. 

#### 3.1.1. Pridine-3-Carbohydrazide (**1**)

Pyridine-3-carbohydrazide was prepared according to previously reported methods [43,58] with some modifications. Briefly, a catalytic amount of 98% sulfuric acid was added to a refluxing ethanolic solution of pyridine-3-carboxylic acid (1 g, 8.1 mmol). Progress of the reaction was monitored on TLC. After the carboxylic acid consumption, the reaction mixture was cooled and neutralized with sodium bicarbonate (10%), followed by the extraction of the product in DCM (3X). The solvent was removed in vacuo to afford a white powdered ethyl ester of pyridine-3-carboxylic acid in quantitative yield. The ethanolic solution of the ester (1.2 g, 8.1 mmol) was then refluxed with five equivalents of hydrazine hydrate (80%). After the overnight reaction, the mixture was cooled, and the solvent was reduced in vacuo, followed by the addition of the mixture on crushed ice. The white precipitates were washed with an excess of cold water and recrystallized in ethanol to get pyridine-3-carbohydrazide as a white powder with an overall yield of 68% (0.74 g). The purity of the product was confirmed with TLC by comparison with a standard product. R_f_: 0.44 (CH_3_OH/CHCl_3_, 1:9); mp: 168–169 °C; ^1^H-NMR (400 MHz, (CD_3_)_2_SO): δ ppm 9.93 (s, 1H, NH), 8.95 (dd, *J* = 2, 0.8 Hz, 1H, CH), 8.67 (dd, *J* = 4.8, 1.6 Hz, 1H, CH), 8.14 (dt, *J* = 7.6, 2.0 Hz, 1H, CH), 7.48 (ddd, *J* = 8.0, 5.2, 0.4 Hz, 1 H, CH), 4.57 (s, 2H, NH_2_); MS (EI) *m*/*z*: 137.1.

#### 3.1.2. Pridine-4-Carbohydrazide (**2**)

Pyridine-4-carbohydrazide was prepared according to previously reported methods [43,58] with some modifications. Briefly, a catalytic amount of 98% sulfuric acid was added to a refluxing ethanolic solution of pyridine-3-carboxylic acid (1 g, 8.1 mmol). Progress of the reaction was monitored on TLC. After the carboxylic acid consumption, the reaction mixture was cooled and neutralized with sodium bicarbonate (10%), followed by the extraction of the product in chloroform (3X). The solvent was removed in vacuo to afford a white powdered ethyl ester of pyridine-3-carboxylic acid in quantitative yield. The ethanolic solution of the ester (1.2 g, 8.1 mmol) was then refluxed with four equivalents of hydrazine hydrate (80%). After the overnight reaction, the mixture was cooled, and the solvent was reduced in vacuo, followed by the addition of the mixture on crushed ice. The white precipitates were washed with an excess of cold water and recrystallized in ethanol to get pyridine-4-carbohydrazide as a white powder with an overall yield of 63% (0.68 g). The purity of the product was confirmed with TLC by comparison with a standard product. R_f_: 0.48 (CH_3_OH/CHCl_3_, 1:9); mp: 170–171 °C; ^1^H-NMR (400 MHz, (CD_3_)_2_SO): δ ppm 10.07 (s, 1H, NH), 8.69 (dd, *J* = 4.4, 1.6 Hz, 1H, CH), 7.71 (dd, *J* = 4.4, 1.6 Hz, 1H, CH), 4.60 (s, 2H, NH_2_); MS (EI) m/z: 137.1.

#### 3.1.3. *N’*-Acetylpridine-3-Carbohydrazide (**3**) 

*N*’-Acetylpridine-3-carbohydrazide (**3**) was prepared by a modified literature method [59]. To a methanolic solution of **1** (0.1g, 0.7 mmol), one equivalent of acetic anhydride was slowly added. The reaction mixture was stirred overnight at room temperature, followed by the evaporation of the solvent in vacuo to get 0.13 g of **3** (white powder). Yield: >99%; mp: 138–140 °C; IR (KBr, cm^−1^): 3283 (m), 2887 (m), 1649 (s, *ν*C = O), 1509 (s, *ν* CNpy), 1418 (m), 1274 (m), 1034 (m), 996( w), 831 (w), 704 (m, *ν* NC_5_H_4_), 633 (m), 515 (m), 447 (w); ^1^H-NMR (400 MHz, CD_3_OD): δ ppm 9.00 (d, *J* = 1.6 Hz, 1H, CH), 8.70 (dd, *J* = 4.9, 1.4 Hz, 1H, CH), 8.27 (dt, *J* = 4.4, 1.8 Hz, 1H, CH), 2.06 (s, 3H, CH_3_); MS (EI) m/z: 179.1, HRMS (EI) *m*/*z* calc. for C_8_H_9_N_3_O_2_ 179.0695; found: 179.0696

#### 3.1.4. *N’-*Butyrylpyridine-3-Carbohydrazide (**4**) 

To a methanolic solution of **1** (0.1g, 0.7 mmol), one equivalent of butyric anhydride was slowly added. The reaction mixture was stirred overnight at room temperature, followed by the evaporation of the solvent in vacuo. The crude product was purified on a silica column using pure ethyl acetate as a mobile phase to get 0.10 g of **3** (white powder) with a yield of 68%. mp: 109–110 °C; IR (KBr, cm^−1^): 3275 (m), 2960 (s, *ν* CH_2_CH_3_), 2872 (m, *ν* CH_2_), 1654 (s, *ν*C=O), 1496 (s, *ν* CNpy), 1464 (m), 1313 (m), 1242 (m), 1220 (m), 1097 (w), 911 (w), 828 (m), 836 (m), 706(m, *ν*NC_5_H_4_), 654 (m), 539 (w), 452 (w). ^1^H-NMR (400 MHz, CD_3_OD): δ ppm 9.00 (d *J* = 1.2 Hz, 1H, CH), 8.71 (dd, *J* = 8.0, 1.2 Hz, 1H, CH), 8.27 (dd, *J* = 8.0, 1.6 Hz, 1H, CH), 7.55 (dd, *J* = 4.8, 1.8 Hz, 1H, CH), 2.99 (t, *J* = 7.2 Hz, 2H, CH_2_), 1.72 (sextet, *J* = 7.6 Hz, 2H, CH_2_), 1.01 (t, *J* = 7.6 Hz, 3H, CH_3_). ^13^ C-NMR (400 MHz, CD_3_OD): δ C 175.24 (C=O), 166.88 (C=O), 153.21 (Ar-CH), 149.35 (Ar-CH), 137.32 (Ar-CH), 130.26 (Ar-C), 125.20 (Ar-CH), 36.73 (CH_2_), 20.01 (CH_2_), 13.93 (CH_3_). MS (EI) m/z: 207.2, HRMS (EI) *m*/*z* calc. for C_10_H_13_N_3_O_2_ 207.1008; found: 207.1014.

#### 3.1.5. *N’*-Hexanoylpyridine-3-Carbohydrazide (**5**) 

To a methanolic solution of **1** (0.1 g, 0.7 mmol), one equivalent of hexanoic anhydride was slowly added. The reaction mixture was stirred overnight at room temperature, followed by the evaporation of the solvent in vacuo. Crude product was purified on silica column using pure ethyl acetate as mobile phase to get 0.17 g of **5** (white powder) with a yield of >99%. mp: 97–98 °C; IR (KBr, cm^−1^): 3209 (m), 2952 (s, *ν* CH_2_CH_3_), 2864 (m, *ν* CH_2_), 1601 (s, *ν* C=O), 1493 (s, *ν* CNpy), 1463 (s), 1254 (m), 1027 (w), 829 (m), 7.03 (m, *ν*NC_5_H_4_), 633(m), 461(w). ^1^H-NMR (400 MHz, CD_3_OD): δ ppm 9.02 (d, *J* = 1.3 Hz, 1H, CH), 8.72 (d, *J* = 4.8 Hz, 1H, CH), 8.28 (d, *J* = 8.0 Hz, 1H, CH), 7.56 (dd, *J* = 7.9, 5.0 Hz, 1H, CH), 2.32 (t, *J* = 7.4 Hz, 2H, CH_2_), 1.70 (quint., *J* = 7.2 Hz, 2H, CH_2_), 1.41–1.37 (m, 4H, CH_2_)_2_), 0.94 (t, *J* = 6.80 Hz, 3H, CH_3_); MS (EI) m/z: 235.2, HRMS (EI) *m*/*z* calc. for C_12_H_17_N_3_O_2_ 235.1321; found: 235.1312.

#### 3.1.6. *N’*-Octanoylpyridine-3-Carbohydrazide (**6**) 

To a methanolic solution of **1** (0.1 g, 0.7 mmol), one equivalent of octanoic anhydride was slowly added. The reaction mixture was stirred overnight at room temperature followed by the evaporation of solvent in vacuo. The crude product was purified on a silica column using pure ethyl acetate as a mobile phase to get 0.17 g of **6** (white powder) with a yield of 87%. mp: 111–112 °C; IR (KBr, cm^−1^): 3254 (m), 2922 (s, *ν*CH_2_CH_3_), 2853 (m, *ν*CH_2_), 1602 (s, *ν* C= O), 1461 (s, *ν* CNpy), 1414 (m), 1278 (m), 1235 (m), 1162 (m), 1025 (w), 870 (m), 699 (m, *ν*NC5H4), 643 (m), 547 (m), 434 (w). ^1^H-NMR (400 MHz, CD_3_OD): δ ppm 9.00 (d, *J* = 1.6 Hz, 1H, CH), 8.70 (dd, *J* = 4.9, 1.4 Hz, 1H, CH), 8.26 (dd, *J* = 6.3, 2.0 Hz, 1H, CH), 7.55 (dd, *J* = 7.9, 4.9 Hz, 1H, CH), 2.31 (t, *J* = 7.4 Hz, 2H, CH), 1.68 (quint., *J* = 7.24 Hz, 2H, CH_2_), 1.38–1.32 (m, 8H, (CH_2_)_4_), 0.90 (t, *J* = 6.8 Hz, 3H, CH_3_); MS (EI) m/z: 263.2; HRMS (EI) *m*/*z* calc. for C_14_H_21_N_3_O_2_ 263.1634; found: 263.1636.

#### 3.1.7. *N’-*Decanoylpyridine-3-Carbohydrazide (**7**) 

*N*’-Decanoylpyridine-3-carbohydrazide (**7**) was prepared by a modified literature method [59]. To a methanolic solution of **1** (0.1 g, 0.7 mmol), one equivalent of decanoic anhydride was slowly added. The reaction mixture was stirred overnight at room temperature, followed by the evaporation of the solvent in vacuo. The crude product was purified on a silica column using ethyl acetate containing 10% methanol as a mobile phase to get 0.13 g of **7** (white powder) with a yield of 61%. mp: 110–112 °C; IR (KBr, cm^−1^): 3195 (m), 2921 (s, *ν*CH_2_CH_3_), 2851 (m, *ν*CH_2_), 1602 (s, *ν* C= O), 1469 (s, *ν*CNpy), 1413 (m), 1225 (m), 1161 (w), 873 (w), 665 (m, *ν*NC_5_H_4_), 550 (w), 442 (w); ^1^H-NMR (400 MHz, CD_3_OD): δ ppm 9.00 (d, *J* = 1.4 Hz, 1H, CH), 8.70 (dd, *J* = 4.9, 1.6 Hz, 1H, CH), 8.25 (dd, *J* = 6.1, 1.9 Hz, 1H, CH), 7.55 (dd, *J* = 7.5, 0.7 Hz, 1H, CH), 2.31 (t, *J* = 7.4 Hz, 2H, CH_2_), 1.68 (quint., *J* = 7.1 Hz, 2H, CH_2_), 1.39–1.30 (m, 12H, (CH_2_)_6_), 0.89 (t, *J* = 7.0 Hz, 3H, CH_3_); MS (ESI) m/z: 367.3; HRMS (ESI) *m*/*z* calc. for C_16_H_25_N_3_O_2_ 292.2020; found: 292.2021.

#### 3.1.8. *N’-*Acetylpyridine-4-Carbohydrazide (**8**) 

*N*’-Acetylylpyridine-4-carbohydrazide (**8**) was prepared by a modified literature method [60]. To a methanolic solution of **2** (0.1 g, 0.7 mmol), one equivalent of acetic anhydride was slowly added. The reaction mixture was stirred overnight at room temperature, followed by the evaporation of the solvent in vacuo to get 0.13 g of **8** (white powder) with a yield of >99%. mp: 161–163 °C; IR (KBr, cm^−1^): 3271 (m), 1623 (s, *ν*C = O), 1512(s, *ν* CNpy), 1413 (m), 1283 (m), 996 (w), 843 (w), 755 (m, *ν* NC_5_H_4_), 596 (m), 460 (w); ^1^H-NMR (400 MHz, CD_3_OD): δ ppm 8.71 (d, *J* = 5.7 Hz, 2H, CH), 7.81 (dd, *J* = 4.6, 1.6 Hz, 2.06 (s, 3H, CH_3_); MS (EI) m/z: 179.2; HRMS (EI) *m*/*z* calc. for C_8_H_9_N_3_O_2_ 179.0695; found: 179.0696.

#### 3.1.9. *N’*-Butyrylpyridine-4-Carbohydrazide (**9**) 

*N*’-Butyrylpyridine-4-carbohydrazide (**9**) was prepared by a modified literature method [60]. To a methanolic solution of **2** (0.1 g, 0.7 mmol), one equivalent of butyric anhydride was slowly added. The reaction mixture was stirred overnight at room temperature followed by the evaporation of solvent in vacuo. The crude product was purified on a silica column using pure ethyl acetate as mobile phase to get 0.10 g of **9** (white powder) with a yield of 67%. Mp: 138–139 °C; IR (KBr, cm^−1^): 3262 (m), 2962 (s, *ν* CH_2_CH_3_), 2668 (m, *ν* CH_2_), 1654 (s, *ν*C=O), 1521(s, *ν* CNpy), 1463 (m), 1310 (m), 1296 (m), 1218 (m), 1065 (w), 846 (m), 836 (m), 749 (m, *ν*NC_5_H_4_), 688 (m), 593(w), 457(w). ^1^H-NMR (400 MHz, CD_3_OD): δ ppm 8.71 (dd, *J* = 4.6, 1.6 Hz, 2H, CH), 7.81 (dd, *J* = 4.6, 1.6 Hz, 2H, CH), 2.29 (t, *J* = 7.3 Hz, 2H, CH_2_), 1.72 (sext., *J* = 7.4 Hz, 2H, CH_2_), 1.01 (t, *J* = 7.4 Hz, 3H, CH_3_). ^13^ C-NMR (400 MHz, CD_3_OD): δ C 175.146 (C=O), 166.76 (C=O), 151.06 (2 × Ar-CH), 141.96 (Ar-C), 123.10 (2 × Ar-CH), 36.71 (CH_2_), 20.00 (CH_2_), 13.92(CH_3_); MS (EI) m/z: 207.0; HRMS (EI) *m*/*z* calc. for C_10_H_13_N_3_O_2_ 207.1008; found: 207.1014.

#### 3.1.10. *N*’-Hexanoylpyridine-4-Carbohydrazide (**10**) 

*N*’-Hexanoylpyridine-4-carbohydrazide (**10**) was prepared by a modified literature method [61]. To a methanolic solution of **2** (0.1 g, 0.7 mmol), one equivalent of hexanoic anhydride was slowly added. The reaction mixture was stirred overnight at room temperature, followed by the evaporation of the solvent in vacuo. The crude product was purified on silica column using pure ethyl acetate as mobile phase to get 0.17 g of **10** (white powder) with a yield of >99%. mp: 99–100 °C; IR (KBr, cm^−1^): 3202 (m), 2950 (s, *ν* CH_2_CH_3_), 2866 (m, *ν* CH_2_), 1604 (s, *ν* C=O), 1550 (s, *ν* CNpy), 1470(s), 1219 (m), 844 (m), 7.56 (m, *ν*NC_5_H_4_), 642 (m), 474 (w). ^1^H-NMR (400 MHz, CD_3_OD): δ ppm 8.73 (dd, *J* = 4.6, 1.6 Hz, 2H, CH), 7.84 (dd, *J* = 4.6, 1.6 Hz, 2H, CH), 2.33 (t, *J* = 7.4 Hz, 2H, CH_2_), 1.71 (quint., *J* = 7.4 Hz, 2H, CH_2_), 1.42–1.37 (m, 4H, (CH_2_)_2_), 0.95 (t, *J* = 7.1 Hz, 3H, CH_3_); MS (EI) m/z: 235.2; HRMS (EI) *m*/*z* calc. for C_12_H_17_N_3_O_2_ 235.1321; found: 235.1312.

#### 3.1.11. *N*’-Octanoylpyridine-4-Carbohydrazide (**11**) 

*N*’-oOctanoylpyridine-4-carbohydrazide (**11**) was prepared by a modified literature method [61]. To a methanolic solution of **2** (0.1 g, 0.7 mmol), one equivalent of hexanoic anhydride was slowly added. The reaction mixture was stirred overnight at room temperature, followed by the evaporation of the solvent in vacuo. The crude product was purified on a silica column using pure ethyl acetate as a mobile phase to get 0.19 g of **11** (white powder) with a yield of >99%. mp: 111–112 °C; IR (KBr, cm^−1^): 3204 (m), 2924 (s, *ν*CH_2_CH_3_), 2855 (m, *ν*CH_2_), 1602 (s, *ν* C= O), 1550 (s, *ν* CNpy), 1404 (m), 1212 (m), 1064 (w), 844 (m), 718 (m, *ν*NC5H4), 679 (w), 639 (m), 444(w). ^1^H-NMR (400 MHz, CD_3_OD): δ ppm 8.71 (d, *J* = 6.0 Hz, 2H, CH), 7.81 (d, *J* = 6.0 Hz, 2H, CH), 2.31 (t, *J* = 7.4 Hz, 2H, CH), 1.68 (quint., *J* = 7.0 Hz, 2H, CH_2_), 1.37–1.32 (m, 8H, (CH_2_)_4_), 0.90 (t, *J* = 6.8 Hz, 3H, CH_3_); MS (EI) m/z: 263.3, HRMS (EI) *m*/*z* calc. for C_14_H_21_N_3_O_2_263.1634; found: 263.1628.

#### 3.1.12. *N*’-Decanoylpyridine-4-Carbohydrazide (**12**) 

*N*’-Decanoylpyridine-4-carbohydrazide (**12**) was prepared by a modified literature method [61]. To a methanolic solution of **2** (0.1 g, 0.7 mmol), one equivalent of hexanoic anhydride was slowly added. The reaction mixture was stirred overnight at room temperature, followed by the evaporation of the solvent in vacuo. The crude product was purified on a silica column using ethyl acetate containing 10% methanol as a mobile phase to get 0.16 g of **12** (white powder) with a yield of 78%. mp: 118–120 °C; IR (KBr, cm^−1^): 3193 (m), 2922 (s, *ν*CH_2_CH_3_), 2853 (m, *ν*CH_2_), 1600 (s, *ν* C= O), 1550 (s, *ν* CNpy), 1468 (m), 1309 (m), 1222 (w), 849 (w), 720 (m, *ν*NC_5_H_4_), 678 (w), 632 (w), 443 (w). ^1^H-NMR (400 MHz, CD_3_OD): δ ppm 8.70 (dd, *J* = 4.6, 1.6 Hz, 2H, CH), 7.81 (dd, *J* = 4.5, 1.6 Hz, 2H, CH), 2.31 (t, *J* = 7.4 Hz, 2H, CH_2_), 1.68 (quint., *J* = 7.12 Hz, 2H, CH_2_), 1.39–1.20 (m, 12H, (CH_2_)_6_), 0.89 (t, *J* = 7.04 Hz, 3H, CH_3_); MS (EI) m/z: 291.2, HRMS (EI) *m*/*z* calc. for C_22_H_29_N_3_O_2_ 291.1947; found: 291.1937.

### 3.2. Antifungal Studies

Three American Type Culture Collection (ATCC) yeast species, including *Candida albicans* ATCC 36082 (fluconazole-resistant), *Candida glabrata* ATCC 2001 (fluconazole susceptible), *Candida parapsilosis* ATCC 22019 (fluconazole-resistant) were obtained from Oxoid, UK and *Candida albicans* CL1 (fluconazole-resistant) was recovered from scalp infection in this study [62,63,64]. Pure cultures were grown onto sabouraud dextrose agar (SDA, Merck, Germany) at 28 °C for 48 h, as described by Pfaller et al. [65]. Standard solutions of 1500 µg/mL of fluconazole (FLZ) (Pfizer, Manhattan, NY, USA) and synthetic compounds were prepared in DMSO (Sigma Aldrich, Germany).

#### 3.2.1. Susceptibility Testing

Susceptibility testing was performed via disk diffusion method at a final concentration of 10 µg/mL for fluconazole and synthetic compounds according to NCCLS guidelines 2004, loaded onto sterile Whatman filter paper disk placed onto the lawns of yeast culture. The zones of inhibitions were calculated in millimeters, and the anti-fungal index (AFI) was estimated as a ratio of inhibitory zones produced by standard concentrations (10 µg/mL) of compounds and fluconazole, respectively.

#### 3.2.2. Anti-Fungal Index (AFI)

The AFI for each compound was calculated by the following formula: 

AFI = Zone of inhibition of compound/Zone of inhibition of fluconazole observed after the 24 h of incubation at 28 °C. AFI values >1 indicate the relative effectiveness of tested compounds against the pathogenic yeast species than the standard concentration of fluconazole used in disk diffusion assay. 

#### 3.2.3. Minimal Inhibitory Concentration (MIC) of Antifungal Agents

The minimal inhibitory concentrations of fluconazole and the synthetic compounds were estimated via the micro dilution method as described earlier [66,67]. Briefly, 100 µL of 1.5 × 10^3^ yeast cells were harvested in 2X concentrated yeast nitrogen base (YNB) broth (Difco, Detroit, Mich.) and added to each microdilution well. 100 µL of standard dilutions of FLZ and compounds (2.0 to 48 μg/mL) prepared in phosphate buffer saline (PBS, pH 7.0) were also dispensed into each well while adjusting the final concentration of DMSO at 5%. Negative control does not supplement with FLZ but comparing the cell suspensions, DMSO and PBS were used to monitor the growth rate. The wells comprising 100 μL of YNB along with 5% DMSO were used as a positive control (neither cells nor antifungal agent was added to the solution). The trays were incubated at 35 °C for 24 to 48 h in ambient air. The plates were then subjected to visual endpoint readings, followed by the OD measurements at 600 nm after 24 and 48 h incubation [62,64]. The MIC was calculated as the lowest concentration of the tested compound, having 80% growth inhibition compared to drug-free growth control [63,66]. 

#### 3.2.4. Time Kill Assay of Antifungal Agents

Compound **6** and fluconazole were evaluated for the time-dependent killing potential against *Candida albicans* ATCC 36082, *Candida glabrata* ATCC 2001, *Candida parapsilosis* ATCC 22019, and *Candida albicans* CL1 through density measurement at 600 nm and viable bacterial cell count during 0–48 h as described earlier [64,65]. Briefly, the strains were cultured to achieve a density of ~10^5^–10^6^ CFU/mL (OD 0.1 @600 nm) in yeast potato dextrose broth (YPD). Aliquots of media containing yeast cells and 0.5X, 1X, and 2X concentrations of MIC (in a final volume of 200 μL in a microwell plate) of the compound and fluconazole were prepared in triplicates. The growth reduction at time intervals of 0, 6, 12, 24, and 48 h of each treatment, along with positive and negative control (medium inoculated with and without viable cells), was observed at 600 nm. Inoculums of 100 µL were withdrawn from each dilution and plated onto yeast potato dextrose agar to determine colony count (CFU/mL) at 37 °C for 48 h. Reduction of < 3 log CFU/mL after the treatment of compounds and fluconazole by virtue of the starting inoculums was defined as fungi static activity; however, reduction ≥3 log CFU/mL was considered fungicidal activity during the assay time proposed by Scorneaux et al. [68].

#### 3.2.5. Hemolytic Assay 

Venus blood (3.0 mL) was drawn in vacutainer tubes (BD) from a healthy individual. The red blood cells (RBCs) were washed twice and re-suspended in phosphate buffer saline (PBS pH 7.4) with a 1:10 ratio. A stock solution of compounds **4** and **6** (2 mg/500 μL) was prepared in DMSO and filtrated through a 0.2 μm syringe filter (Corning, Germany). Standard solutions of ampicillin/cloxacillin (2 mg/500 μL) and fluconazole (2 mg/1000 μL) prepared in sterile water. The stock solutions of drugs, RBCs, DMSO (1–5%), PBS (negative control), triton X100, and SDS (1%, positive control) were preincubated at 37 °C before performing the hemolytic assay. Aliquots of 190 μL PBS diluted blood were dispensed into the micro-titer plates in duplicate at 37 °C. RBCs were treated with a two-fold rise (4–32 μg) in the concentration of compounds **4** and **6,** ampicillin/cloxacillin (32 μg), and fluconazole (32 μg) up to 30 min at physiological temperature. Blood cells incubated with 1% (*w*/*v*) triton X-100, SDS, DMSO (1–5%), and PBS were considered positive and negative controls, respectively. The amount of hemoglobin released from the ruptured RBCs was estimated at 576 nm and transformed into the percent hemolysis [69].

### 3.3. Antibacterial Studies

Six Gram-negative (Aeromonas hydrophila ATCC 7966; Proteus mirabilis ATCC 12453 and 212a (isolated from the soil); Pseudomonas aeruginosa ATCC 27853 and PA-01; Salmonella typhi XDR ST-CL-15 (isolated from blood culture) and two-Gram positive (Enterococcus faecalis ATCC 29212; Staphylococcus aureus ATCC 29213) cultures were obtained from ATCC and indigenous clinical specimens. ATCC and two indigenous bacterial strains included in this study harbor drug resistance against any antimicrobial classes, including the cell wall, protein, and DNA synthesis inhibitors, respectively [70]. Aeromonas hydrophila ATCC 7966; Proteus mirabilis ATCC 12453, and Enterococcus faecalis ATCC 29212) are resistant to more than one antibiotic [57,70,71]; therefore, we included them in our study. 

#### 3.3.1. Susceptibility Profiling of Antibacterial Compounds

For the synthesized compounds, antimicrobial susceptibilities were tested by disk diffusion assay using ampicillin/cloxacillin as a standard antimicrobial agent [72]. Briefly, the broad-spectrum antimicrobial activity was observed on confluent lawns of bacterial cultures having 1.3 × 10^8^ cells per milliliter. The standard solution (32 μg/mL) of the compounds was prepared in DMSO (5% *w*/*v*) through filtration using a 0.2 μm syringe filter (Corning, Germany). Sterile paper disks (Whatman filter paper #1) of 5.0 mm diameter (1.0 mm thickness) were impregnated and placed over the lawn of cultures along with ampicillin/cloxacillin disks (32 μg/mL) as control. The zone of inhibitions (in millimeters) was measured after 24 h incubation at 37 °C. The susceptible bacterial species were then evaluated for the determination of minimal inhibitory concentration and a time-kill assay of the promising compounds.

#### 3.3.2. Minimal Inhibitory Concentration (MIC) of Antibacterial Compounds

Minimal concentrations of drug at which bacteria are susceptible to successful treatment with an antibiotic or a compound were determined via the micro broth dilution method described earlier [55,72]. 

#### 3.3.3. Time Kill Assay of Antibacterial Compounds

Time kill assay against the *Pseudomonas aeruginosa* PA-01 and *Staphylococcus aureus* ATCC 29213 was evaluated for the time-dependent killing potential of compound **4** and ampicillin/cloxacillin through optical density (OD) measurement at a wavelength of 600 nm and viable bacterial cell count at different time intervals as described earlier [50]. Briefly, bacterial strains were cultured to achieve a density of ~10^6^ CFU/mL (OD 0.09@600 nm) in Mueller Hinton Broth (MHB). Aliquots of media containing bacterial cells and 0.5 X, 1X, and 2X concentrations of MIC (in a final volume of 200 μL in a microwell plate) of compound **4** and ampicillin/cloxacillin were prepared in triplicates. The growth reduction at time intervals of 0, 6, 12, 24, and 48 h of each treatment, along with a positive and negative control (medium inoculated with and without viable cells), was observed at 600 nm. Inoculums of 100 µL were withdrawn from each dilution and plated onto nutrient agar to determine colony count (CFU/mL) at 37 °C for 24 h. The time-kill measurement and the rate of bacterial death were determined by plotting the log10 (CFU/mL) against the time. 

### 3.4. Computational Studies

In order to reveal the binding mode and inhibition mechanisms of the newly synthesized pyridine-containing lipophilic hydrazides for the antibacterial and antifungal activities, molecular docking was performed against the well-established targets CYP51 *Candida glabrata* (PDB ID 5JLC) as fungal target and topoisomerase IV (PDB ID 3FV5) as bacterial target [56]. The crystal structures of the target proteins were retrieved from Protein Data Bank (www.rcsb.org/pdb). All the proteins were prepared, protonated, charged, and minimized via MOE 2019 suit. The chemical structures of the potentially active synthesized derivatives were constructed by the Builder tool integrated into MOE and saved their 3D conformations. Further structure preparations include atom-type corrections, protonation, and minimization. Charge applications were also made by MOE. Prior to docking, redocking was performed to assess the efficiency of the docking protocol and software. Redocking results confirmed MOE as an appropriate software for docking these hydrazide-based lipophilic compounds.

## 4. Conclusions

Contemporary antimicrobial drug discovery necessitates the determination of small molecule targets, particularly for microorganisms that are multidrug resistant. This work allows us to rapidly generate a library of pyridine carboxylic acid hydrazides of lipophilic nature. The intriguing structural importance of pyridine coupled with hydrazide moiety and aliphatic chains rendered their potency against diverse microbial strains. Specifically, we were able to devise a connection between diverse structures of altered hydrazide derivatives along with their antimicrobial profile against multidrug-resistant (MDR) strains. In addition, computational studies provided useful information on their interaction with target proteins. Overall results suggest that this type of compound possesses few harmful effects and has the potential to be promising drugs against MDR strains of bacteria and fungi. 

## Data Availability

See Appendix A.

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
