# Peer review of "Designing Functionally Substituted Pyridine-Carbohydrazides for Potent Antibacterial and Devouring Antifungal Effect on Multidrug Resistant (MDR) Strains"

_molecules, 2022, doi:10.3390/molecules28010212_

Round 1

Reviewer 1 Report

The manuscript is decent but the following points need to be addressed before it can be accepted for publishing:

1.  What is the MIC value of the most active compound from Ramani et al. (line 128-129)? It is suggested to include the MIC / IC50 value of the most active molecule.

2.  For anti-bacterial molecular docking study, the rational for choosing topoisomerase IV (PDB ID 3FV5) need to be detailed clearly.

3.   The melting point of intermediates and final compounds is missing and it needs to be incorporated.

4.    All the scientific names need to italicized throughout the manuscript. (For eg., in line 325, P. aeruginosa, P. mirabilis, S. aureus etc. need to be italicized).

5.   As most of the acid hydrazides have decent anti-tubercular activity, it is suggested to screen the synthesized derivatives against resistant TB strains as well and include the results in the manuscript.

Author Response

Enclosed file

Reviewer 2 Report

Dear Authors,

The manuscript ID: molecules-2059613-v1 entitled Designing Functionally Substituted Pyridine-Carbohydrazides for Potent Antibacterial and Devouring Antifungal Effect on Multidrug Resistant (MDR) Strains” is interesting.

In the last years, the antimicrobial resistance has become very widespread and is also one of the most serious global public health threats. Therefore, it is necessary to search for new antibacterial and antifungal drugs. The whole manuscript is properly organized. The results are documented, summarized in the form of schemes, tables or figures and interpreted. Based on the results, discussion and conclusions were drawn.  

However, the agar well diffusion method and determination of the zones of inhibition of microbial growth by these compounds are only preliminary and screening tests. Moreover, according to European Committee on Antimicrobial Susceptibility Testing (EUCAST) and National Committee for Clinical Laboratory Standards (NCCLS) guidelines, to verify the antifungal activity, RPMI 1640 (with MOPS) medium should be used. Moreover, the Authors presented antibacterial and antifungal activity as MIC. There is no information about the MBC (Minimal Bactericidal Concentration), MFC (Minimal Fungicidal Concentration), and MBC/MIC or MFC/MIC values and bactericidal/fungicidal or bacteriostatic/fungistatic effect) of the studied compounds. Publication would be more valuable with these additional informations. In the next work, please remember this.

I have some suggestions in order to improve this paper, which are the following:

1)   Introduction: Please clearly state the purpose of the study;

2)   Methods:

- „100 μL of standard dilutions of FLZ and compounds (2.0 to 48 μg/mL) in phosphate buffer saline (PBS, pH 7.0) were also dispensed into each well, while adjusting the final concentration of DMSO at 5%.” – Standard dilutions of FLZ and compounds should be prepared in liquid medium instead of PBS (when adding them directly to the wells of the plate)

- co-amoxicillin ??? Ampicillin + cloxacillin – which antibacterial drugs were reference in this study?

- What was the purpose of the hemolytic study? What ranges of haemolytic action are acceptable for drugs?

3)   Nomenclature of microorganisms (with italics), if a microorganism is mentioned again in the text, please write its name in the abbreviated version, i.e. Candida albicans – C. albicans, etc.

4)   Anti-fungal index (AFI) – please add literature data to this study;

5)   Standard dilutions of FLZ and compounds should be prepared in liquid medium instead of PBS (when adding them directly to the wells of the plate);

6)   The results should be sorted:

- „Compound 4, 5, 8 – 10 and 11 inhibited the growth of Proteus mirabilis 212a” – Table 2: is compound 12 instead 11

- „Similarly, minimum concentration of 4 μg/mL of compound 4 and 9 was required to inhibit the growth of Pseudomonas aeruginosa ATCC 27853 and PA-01” – Table 2: compound 9 is inactive against PA-01;

- „The lowest MIC value of 2 μg/mL was observed for …….. and compound 6 against Staphylococcus aureus ATCC 29213” and „Similarly, compound 6 significantly inhibited the growth of S. aureus ATCC 29213 at 2 μg/mL” – the same information, please corect this text

7) other:

- throughout the text – please standardize: the spacing between words, „Pridine” – Pyridine;

- Line 669: stains – strains

I think, taht the obtained results are valuable and manuscript is worth publishing in „Molecules”, after major review.

With highest regards,

Author Response

Enclosed file

Reviewer 3 Report

In the manuscript ID: molecules-2059613 the authors describe the development and testing of pyridine-carbohydrazides as antifungal and antibacterial compounds. Specific hits in the obtained molecule series, characterized by various lipophilic chains of different lengths, showed a good antimicrobial activity, both against Candida spp. (compound n. 6) and different bacterial isolates (compound n.4), which was greater than that of reference drugs adopted against these microorganisms.

The paper sounds interesting, with a detailed and comprehensive background section, allowing the reader to understand the topic. However, while the chemistry and mycology sections are fine and well commented, there are some concerns in the bacteriological experiments that need to be discussed:

-The authors state they have performed the assays on multidrug-resistant strains; however, the used bacterial strains are reference ones, usually adopted as controls in antibiotic susceptibility assays and known as susceptible. Could they please provide evidences of the stated antibiotic resistance traits (i.e.; Kirby-Bauer or Minimum Inhibitory Concentrations, MICs, results for at least three antibiotics belongings to different classes)?

-Table 2 reports the MIC values for the ampicillin/cloxacillin combination; however, the results should be expressed for both the antibiotics in the combination, as done for the β-lactams/β-lactamase inhibitors associations, and not as a single value. Moreover, ampicillin is not usually adopted against Pseudomonas aeruginosa and the results obtained for Staphylococcus aureus and Enterococcus faecalis exceed the recommended range of values for ampicillin alone. Can the authors comment on this discrepancy?

-At the end of the microbiological section the authors suggest that the compound n.4 could act by impairing the bacterial membranes, while in the subsequent molecular docking section they assess the compound binding to the bacterial topoisomerase IV. Can the authors explain this choice of changing target?

Moreover, no mention of statistical analysis (especially in time killing assays and hemolysis tests) is provided and figures lack standard deviation.

Based on these observations, the paper needs major revisions before being published in “Molecules”.

MINOR COMMENTS

Please type the microorganism species names in italic throughout the manuscript;

Line 71, please correct “Proteus mirabilis”;

Line 225, please type “in vitro” and “in silico” in italic;

Line 631, please correct “in aerobic environment”;

Please unify the two “Hemolytic assay” section in the methods section;

Please provide better quality supplementary material.

Author Response

File attached

Round 2

Reviewer 1 Report

Extensive revision is done and the article can now be accepted for publication.

Author Response

attach filed

Reviewer 2 Report

Dear Authors,

 The manuscript ID: molecules-2059613-v2 entitled Designing Functionally Substituted Pyridine-Carbohydrazides for Potent Antibacterial and Devouring Antifungal Effect on Multidrug Resistant (MDR) Strains” was corrected according to review’s suggestions. I have no other comments on this article. I think that manuscript is worth publishing in Molecules.

With highest regards,

Author Response

attach file 

Reviewer 3 Report

In the revised version of the manuscript ID: molecules-2059613 the authors have addressed most of the raised critics, improving the quality of the paper. There are still two concerns that should be discussed:

-Regarding the antibiotic susceptibility values, the MICs reported for Pseudomonas aeruginosa ATCC 27853 by Sekiguchi et al. are actually 4 and 32 µg/ml for piperacillin-tazobactam and sulfamethoxazole-trimethoprim, respectively, which are in line with the reference guidelines (CLSI 2017) but do not imply antibiotic resistance. For example, piperacillin-tazobactam resistant strains exhibit a MIC value higher than 16 µg/ml. For kanamycin and cefoxitin there are no reported breakpoints or quality control ranges in reference guidelines, as these antibiotics are not usually adopted against P. aeruginosa, thus assessing the related resistance is debatable.

According to this reviewer, it is important to underline the effectiveness of the described compounds against both reference and indigenous strains, but it is improper to refer to them as MDR strains; such correction is advised.

-When determining MIC values, standardized planktonic cultures are directly exposed to the antimicrobials, thus biofilm formation should be excluded and cannot be considered as explanation for high values in Staphylococcus aureus and Enterococcus faecalis strains. The corresponding sentence should be deleted, while the choice of the association ampicillin/cloxacillin can be accepted, even if a more suitable broad-spectrum antibiotic (e.g., ciprofloxacin) is advised as reference drug.

After performing these minor revisions, the paper can be accepted for publication.

Author Response

enclosed file
